# Creep Life Prediction of 10CrMo9–10 Steel by Larson–Miller Model

**DOI:** 10.3390/ma15134431

**Published:** 2022-06-23

**Authors:** Agnieszka Zuzanna Guštin, Borut Žužek, Bojan Podgornik

**Affiliations:** Institute of Metals and Technology, Lepi pot 11, 1000 Ljubljana, Slovenia; borut.zuzek@imt.si (B.Ž.); bojan.podgornik@imt.si (B.P.)

**Keywords:** Larson–Miller extrapolation model, creep life modeling, experimental data, time-temperature parameter *C*

## Abstract

Creep is defined as the permanent deformation of materials under the effect of sustained stress and elevated temperature for long periods of time, which can essentially lead to fracture. Due to very time-consuming and expensive testing requirements, existing experimental creep data are often analyzed using derived engineering parameters and models to predict and find the correlations between creep life (time to rupture), temperature and stress. The objective of this study was to analyze and compare different numerical algorithms by using the Larson–Miller parameter (LMP) extrapolation model. Calculations were performed using the classical LMP equation where different values of parameter C were selected, as well as using a modified LMP equation in which parameter C was stress dependent C(σ). The impact of two different approaches of extrapolation and correlation functions (linear and polynomial) applied to fit the LMP model was also investigated. A detailed analysis was performed to choose the best extrapolation fit function and error tolerance. The numerical algorithm implemented was validated through creep rupture testing performed on 10CrMo9–10 steel at 600 °C (873 K) and 80 MPa. Creep model behavior analysis proved that different values of C can significantly change the estimated time to rupture. An excellent response of the LMP model was obtained by considering polynomial dependency when parameter C was assumed to be 18, especially for the temperature range from 773 to 873 K. Promising results were also achieved when parameter C was taken as stress-dependent, but only for linear fitting, which requires further analysis. However, at validation stage it turned out that only the linear extrapolation function and C taken as a constant value provided adequate time-to-rupture prediction. In the case of C = 18, estimated time was slightly overestimated (~8%) and for C = 20 it was underestimated by 27%. In all other cases error largely exceeded 50%.

## 1. Introduction

Creep is defined as the permanent deformation of materials under the effect of sustained stresses and elevated temperatures for long durations of time, which can essentially lead to fracture [1]. A comprehensive study and analysis of a material’s behavior under different loading conditions should be made before the material is considered for a particular application. For this purpose, extensive creep testing schemes are carried out at different stress levels and temperatures in order to provide the necessary information regarding long-term material behavior. However, these testing and programming schemes are very expensive and, above all, time-consuming.

Results of the existing short-term measurements can be extrapolated to the loading conditions (stress, temperature) of interest by using creep parametric methods. Existing experimental creep data are analyzed using derived parameters in order to find a correlation between creep life (either time to rupture) and applied stress or temperature. Different extrapolation techniques have been developed [2] over the years for the purpose of predicting the long-term creep behavior of materials without the need to carry out practical tests, which could take many years before being able to design and manufacture the required component. Thus, parametric methods play a key role during the design stage.

Many techniques and approaches have been proposed [3] in an effort to predict long-term creep properties and to reduce the time and costs required to obtain such long-term data. Mostly they are based on the extrapolation of short-term creep-rupture data using time–temperature parametric relationships and calculation of master curves. Based on the calculated master curves the extrapolation to longer times can then be easily obtained. However, this concept is based on the assumption that all creep-rupture data for a given material can be correlated to a single “master curve” wherein the stress (or log stress) is plotted against a parameter involving a combination of time and temperature.

Common methodologies suggested by researchers to predict the creep life of components are Larson–Miller [4], Orr–Sherby–Dorn [5] and Mansen–Haferd [6] parametrization models. Those parametric models have the great advantage of requiring only a relatively small amount of data to establish the required master curve. Among those parametric techniques, the Larson–Miller parametric model (LMP) [7,8] for extrapolating stress-rupture data is chosen mainly due to its simplicity and as it is the most commonly used model to predict the creep life of materials.

The first part of this article investigates the efficiency of the model by matching the fitting material parameter. The calculations are performed either by the classical equation where different values of parameter C are directly selected or by a modified equation, where parameter C is stress dependent C(σ). The calculations performed using the two LMP equations are briefly compared and discussed. The second part of the article investigates the impact of different approaches in terms of extrapolation and correlation functions applied to fit the equations. A detailed analysis is performed to determine the best master curve and error tolerance.

Extrapolation methods are the most widely used models for predicting the creep life of alloys. They can be applied very easily and straightforwardly as soon as the required fitting-parameters are properly selected. However, those parameters may depend on many factors, e.g., the initial microstructure, grain size, chemical composition and deformation rate. Therefore, selecting constant values can lead to misleading results. Furthermore, when selecting fit functions, there is always a question as to which one is the most suitable for a specific temperature range. Therefore, the main aim of this study was to provide an overview of the influencing components for applying the LMP extrapolation model, the difference between various fit functions (linear and polynomial) and constant values, and how to obtain reliable assessments. The emphasis and contribution of this research show that finding the best-fit curve is not always an easy task, especially when variations in the data and parametric constants require different fittings. In practice the user should examine the effect of different fitting functions on the extrapolated values as well as including sufficiently representative data over a wide range of test conditions to select the appropriate function.

The approach and the numerical algorithms implemented were evaluated through the comparison of creep life assessment and real creep rupture time of 10CrMo9–10 steel, determined by a uniaxial creep rupture test conducted at an applied stress of 80 MPa and temperature of 600 °C (873 K). The correlation between Celsius [°C] and Kelvin [K] follows the equation T [K] = 273 + T [°C].

## 2. Creep Modeling

### 2.1. Model Design and Specification

The LMP extrapolation model is one of the most commonly applied extrapolation methods to determine the stress-rupture behavior in metals [9,10,11,12,13]. The traditional approach of creep life prediction is based on a power law relationship between temperature *T* and stress *σ*. The classic power law equation, which is a combination of Arrhenius and Norton [14] equations, is the most established description of the creep of material. The equation is defined as
(1)$$\dot{\varepsilon }=A\;\sigma^{n}e^{\left(-Q/RT\right)}$$
where $\dot{\varepsilon }$, A, σ*,*
n, Q and *R* represent minimum creep rate, material parameter, applied stress, stress exponent, the activation energy for creep and the universal gas constant, respectively. The Monkman–Grant equation [15] is defined as
(2)$${\dot{\varepsilon }}^{\alpha }t=C$$
where t, C and α represent the time to rupture [h], Larson–Miller constant, which is a material-dependent value, and the slope of $\log t\;\mathrm{vs}.\;\log\dot{\varepsilon }\;$(with α=1), respectively. Combining Equations (1) and (2) and taking the logarithm of both sides, the following correlation is obtained
(3)$$\frac{Q}{R}=T\left(\log t+C\right)$$

Equation (3) is usually written as
(4)P=f(σ)=T (log t+C)
where P is the Larson–Miller parameter, defined as a function of stress and referred to as a master curve. The master curve can be obtained for any creep measured values, where in general, only a small range of measured data are needed for extrapolation purposes.

The LMP model includes material parameter C, which is considered to be a constant in almost all cases. Larson and Miller proposed that the C value could be taken as 20 for many metallic materials [16]. However, it was found that the value of this constant varies from one alloy to another and is also influenced by factors such as cold-working, thermo-mechanical processing, phase transitions and/or other structural modifications. Furthermore, calculations also indicate that the creep life estimation is very sensitive to small differences in C.

C is obtained mostly from experimental creep testing data and is determined as the intersecting point of iso-stress lines with the *y*-axis on a log *t* vs. 1/*T* plot [16]. Moreover, most applications are made by first calculating the C parameter that provides the best fit to the data, which means that C is treated as a fitting value. The LMP model is used under the assumption that the iso-stress lines, when extended, cross the vertical axis (at 1/*T* = 0) in a single point. However, as is evident from Figure 1, iso-stress plots intersect the vertical axis in some range depending on the applied stress. According to this, and as proposed by A. Ghatak et al. [17], value C can also be taken as dependent on stress. Equation (4) can therefore be modified in the form considering the value C as the function of stress (C(*σ*)) and not as a constant value [18,19]:(5)$$P=f\left(\sigma \right)=T\;\left(\log\;t+C\left(\sigma \right)\right)\cdot 10^{-3}$$

### 2.2. Creep Data

The material chosen for this study was 10CrMo9–10 steel (nominal composition in wt.%: 0.05–0.15% C–Carbon, <0.5% Si, 0.30–0.60% Mn–Manganese, 2.0–2.5% Cr–Chromium, 0.90–1.15% Mo–Molybdenum), which is the most commonly employed material in power generating and refining plants. In order to perform the creep-damage calculations, the extrapolation algorithm requires creep testing data extracted from real sets of creep tests. In this study, data were obtained from an available existing on-line creep database from the National Institute for Materials Science of Japan [20]. The database includes the creep–rupture testing results for standard creep-rupture tests performed at different temperatures and at various stress levels. Every test was performed at a single temperature and at a single constant load up to the rupture.

For the model design purpose, the set of twelve creep tests of 10CrMo9–10 steel were selected [20]. The selected creep tests were performed at specified conditions, including three different stress levels (150, 200 and 250 MPa) and temperatures ranging from 723 to 873 K. For each stress level, four creep tests at different temperatures were executed. The creep testing data used (log *t* vs. 1/*T*) are plotted in Figure 1.

### 2.3. Model Validation

The task of the validation process was to confirm that the outputs of the investigated LMP creep-model were acceptable and accurate with respect to the real creep-rupture testing. Therefore, the uniaxial creep rupture test was conducted using a ZWICK high-temperature testing machine at 600 °C (873 K) with applied stress of 80 MPa. Creep test specimens with a diameter of 4 mm were machined from 10CrMo9–10 steel tube as depicted in Figure 2. Creep rupture testing was conducted up to the rupture of the specimen.

### 2.4. Statistical Comparison

Extrapolation methods and algorithms applied are compared with the fit adequacy of master curves for comparison sets [21]. Each comparison set represents values of measured time to rupture $t_{M}$ and estimated time to rupture t, calculated by the LMP extrapolation. $E\left(t_{M}\right)$ is the mean value of experimental time to rupture for each comparison set. The residual sum of squares represented by Equation (6) is compared to the total sum of squares Equation (7), following the equations:(6)$$RSS=\sum_{i=1}^{j}{\left(t_{M\;i}-t_{i}\right)}^{2}$$
(7)$$TSS=\overset{j}{\underset{i=1}{\sum }}{\left(t_{M\;i}-E\left(t_{M}\right)\right)}^{2}$$
where $t_{M\;i}$ represents the measured time to rupture in [h] and t represents the estimated time to rupture for definite set–in [h]. Finally, the fit adequacy M can be introduced by comparing RSS and TSS:(8)$$M=1-\frac{\sum_{i=1}^{j}{\left(t_{M\;i}-t_{\;i}\right)}^{2}}{\sum_{i=1}^{j}{\left(t_{M\;i}-E\left(t_{M}\right)\right)}^{2}}$$

If the M value is near 1 the model seems to be acceptable for that comparison set. If it is 0 or even negative the model appears to be useless in that particular area.

## 3. Solution Procedures

In this section, the solution procedure providing the estimation of time to rupture t for the selected 10CrMo9–10 steel is described. The parameter *P* from Equation (4) is a function of stress f(σ) and is referred to as a master curve, which can be obtained for any creep measured values by using the LMP model. As already mentioned, for the extrapolation purpose only a small range of measured data is needed and thus considered. In this study, two correlation functions have been used, linear and polynomial, following the equations:(9)P=f(σ)=Eσ+F
(10)$$P=f\left(\sigma \right)=E\sigma^{2}+F\sigma +G$$
where the coefficients *E*, *F* and *G* are obtained from the fit of the plot.

In terms of material parameter C from Equation (4), two constant values are considered, C = 18 and C = 20, and, as proposed by A. Ghatak et al. [17], also taken as dependent on stress Equation (5). In the current case the simplest form of linear dependency is taken into account:*P* = *T* ((log *t* + (*A*_1_·*σ* + *A*_2_))·10^−3^(11)

The coefficients *A*_1_ and *A*_2_ are obtained from the linear fit of plot C versus stress *σ* (see Figure 3), with the value C for the investigated material and testing conditions ranging from 10 to 20 (Figure 1). From Figure 3 it can be observed that for the given stress range the value of C varies linearly with stress following the equation:(12)C(σ)=−0.08112 σ+30.48

In view of the low amount of data (only three stress levels), the linear correlation function was used in further calculations. It would be also interesting to verify other extrapolation functions; however, this is out of the scope of this investigation.

For all assumptions of parameter C the calculated time to rupture follows the equation
(13)$$\log t=P\cdot 10^{3}/T-C$$
with C taken as a constant (C = 18 or C = 20) or function of stress Equation (12). During the discussion, the computed time to rupture will be presented as log *t* for easier data comparison.

## 4. Results and Discussion

The computing results of Larson–Miller parameter P by considering parameter C as a constant value and as a function of stress C(σ) are depicted in Figure 4.

The linear Equation (9) and polynomial Equation (10) extrapolation functions were selected to fit the data. By comparing three charts drawn in Figure 4, the best fit was obtained when considering C as a function of stress (Figure 4c), followed by C taken as constant value of 18 (Figure 4a), and a constant value of 20 (Figure 4b) giving the weakest fit. Selecting linear master curve Equation (9) gave coefficients of determination (R^2^) [22] of 0.98, 0.88 and 0.79 by considering C as stress dependent, 18 and 20, respectively. Very similar values of 0.99, 0.89 and 0.79 were found for the master curve taken as polynomial function Equation (10).

The parameter P can be easily determined for any stress values taking different approaches of C. Table 1 gives the results for various P according to the specified stress.

According to the time to rupture calculation procedure (Equation (13)), the computed time t was compared with the available creep–test data [18], as described in Section 2.1. The results in log *t* form are graphically presented in Figure 5 and Figure 6. For better visualization the calculated time to rupture values are also presented as the percent error (%) and listed in Table 2.

As is generally known and shown in Figure 5 and Figure 6, creep life decreases with an increase in temperature and/or stress. However, the following summary can be drawn by comparing the modeling results with the creep testing data.

Creep model behavior and comparison with the available creep test results when considering the linear master curve (Equation (9)) are depicted in Figure 5. When the calculations are performed by assuming C as a constant value of 20 (Figure 5b), the error starts to increase as we move toward low temperature values. For stress of 250 MPa and temperature of 723 K, the predicted rupture time (log *t* = 4.07) is considerably overestimated with respect to the creep testing data (log *t* = 3.35), while it is substantially underestimated for a high temperature of 873 K (log *t* = −0.06 vs. log *t* = 0.51). Good calculations are obtained in the mid temperature range 773–823 K. The estimated log *t* values for 823 K are 2.80 (150 MPa), 1.98 (200 MPa) and 1.14 (250 MPa) and the corresponding creep testing values are 3.01 (*σ* = 150 MPa), 2.10 (*σ* = 200 MPa) and 1.45 (*σ* = 250 MPa). The percentage error in the mid temperature range is between 14 and 53% and extends to about 75% for high-temperature (873 K) and even over 400% for low-temperature values (723 K). Much better results and a good model response are achieved when parameter C is assumed to be 18 (Figure 5a). Noticeable deviations are observed only when calculations are performed at a low temperature of 723 K and sustained stress values (200 and 250 MPa), with the error exceeding 200%. Otherwise, error is within 10–60%, depending on the stress level, as shown in Figure 5a and Table 2. The best results and the smallest errors are obtained when performing calculations by considering C as a function of stress (Figure 5c). A very good response is observed in almost all of the temperature and stress ranges analyzed.

Creep model behavior compared with creep test results when selecting the polynomial correlation function for the LMP master curve are depicted in Figure 6. In this case, calculations in general give overestimated results, especially if C is taken as stress-dependent (Figure 6c). When C is assumed to be 20 (Figure 6b), the results are comparable to the linear extrapolation, with highly overestimated values for low temperature 723 K. In this case, for a stress level of 150 MPa, the computed log *t* value is 6.1 (measured log *t* = 5.7), for stress level 200 MPa log *t* is 5.08 (measured log *t* = 4.3) and for stress level 250 MPa log *t* is 4.3 (measured log *t* = 3.3). Error is over 150% and increases with increased stress level. However, at 823 K a very good fit is obtained, being within 10–15%. Overestimated values are also observed when parameter C is taken as 18, but with the smallest error (1–6% for 823 K and 11–36 for 873 K), as shown in Figure 6c. The analysis indicates that when using a polynomial correlation function and constant values of parameter C, a good model prediction can be expected for temperatures higher than 723 K. However, again, a certain deviation from the measured values is observed, especially for the highest stress level.

It must be emphasized that extrapolated results in terms of estimated creep life are very sensitive to small differences in temperature and C value, even if C is considered as a constant. Master curves generally gained by the different time–temperature parameter models may look very similar at first, but a detailed analysis (using fit adequacy parameter M; Equation (8)) reveals a significant difference in assessment of the creep rupture time. Assessment of the model parameters represents a significant challenge regardless of which correlation should be used. At this point it must be emphasized that the value of the fit adequacy parameter M cannot be used to predict whether or not a model is good. However, it gives an insight into which model specifications are better than the others.

Based on the calculated M values, a good fit and the most accurate results are obtained when considering the linear correlation function (Figure 7a). When C is taken as a constant of 18, comparable results between model and experimental data are obtained for almost the whole temperature range. The M values range from 0.98 to 0.74 ($M_{723}\;$= 0.82, $M_{773}\;$= 0.98, $M_{823}\;$= 0.92, $M_{873}\;$= 0.74). Even better comparability can be observed for the stress-dependent parameter C, with M being over 0.9. However, a considerably reduced fit (M < 0.6) for the boundary temperatures (723 and 873 K) is obtained when considering C = 20. The use of the extrapolation curve where data are fitted by the polynomial function is reliable only for a certain temperature range and only when considering material parameter C as a constant, as shown in Figure 7b. The best results are obtained when C as constant values of 18 and 20 are used. For C = 18 from 773 K to 873 K, the value of M is over 0.92. For C = 20 a good fit (M > 0.85) is obtained only for a temperature range of 773–823 K. However, when C is assumed as stress-dependent, the computed values are largely overestimated. In this case, the values of fit adequacy parameter M are zero for all data sets, indicating largely questionable validity of the model.

Finally, the validation process of the creep-model was performed by comparing the LMP-model-estimated creep-rupture time with the real, experimentally obtained time. The creep test was performed at a temperature of 600 °C (873 K) and a stress level of 80 MPa. The creep-test result in the form of strain vs. time curve is shown in Figure 8. As shown, the time to rupture was about 560 h. The elongation A = 29.6% and contraction Z = 88.7%

The estimated rupture times can be obtained by considering different fit functions and C values as already described in the previous section. Results for the estimated time to rupture, predicted for the test conditions of 873 K and 80 MPa, are shown in Table 3. It can be observed that the LMP creep-model gives the best behavior and estimation when the linear function extrapolation is applied. For C = 18, estimated time to rupture is about 8% overestimated; it is 27% underestimated for C = 20 and about 65% underestimated when C is taken as a function of stress. However, polynomial function extrapolation gives massively (more than 4 times) overestimated results (Table 3).

Based on the adequacy parameter M analysis, the best result should be achieved by considering C as a constant value of 18 and the polynomial dependency following the equation $P=5\times 10^{-5}\sigma^{2}-0.0326\sigma +21.059$. In this case, the adequacy parameter M is 0.93 and the error between measured and calculated values (*σ* = 150 MPa, T = 873 K) ~35%. Nevertheless, the computed result for 837 K and 80 MPa is 3174 h, which is considerably higher than the experimental result.

However, when using the linear fit function P=−0.0138σ+19.255 , where the reliability of adequacy parameter M is 0.84 but the error at 150 MPa and 873 K is just 3%, the estimated time to rupture at 80 MPa is 618 h, indicating a great model response.

These results clearly demonstrate that the M value cannot be used to predict whether or not a model is appropriate. It shows an insight into which model specifications are better than the others. However, in the current case, the validation of the model requires additional sustained stress creep-testing to be performed to confirm which extrapolation function is more reliable.

In the future, in general, the important topic of this study would be the selection of the input data required for LMP extrapolation model fitting, which was not described in the article, but definitely requires additional analysis and discussion. A particularly important problem in creep-model design is making a choice about the appropriate amount and relevance of the input data. This choice may not be correct if made only on short-term testing and limited testing conditions. The relative strength of materials can change with time. Furthermore, relatively small increases in stress and/or temperature can drastically change material lifetimes.

## 5. Conclusions

Applicability of the LMP extrapolation model by applying different correlation function algorithms and parameter C values has been analyzed and results validated through creep rupture testing of 10CrMo9–10 steel at sustained stress levels. The main conclusions can be summarized as follows:

Results show that different values of C can significantly change the estimated time to rupture. In the case of 10CrMo9–10 steel, better results were obtained when considering C = 18, especially for a higher temperature range of 773 to 873 K. Model prediction error in this temperature range was about 25% (for comparison, when C = 20 it was about 40%). Overall, a poor model response was observed for a low temperature of 723 K, regardless of the parameter C value selected.

Additionally, there was a visible impact of the fitting function on the rupture time calculations. An excellent response of the model was observed by considering polynomial dependency, especially when parameter *C* was assumed to be 18. For the temperature range 773 to 873 K and lower stress levels (<250 MPa), model error dropped even below 20%. However, it rapidly started to increase at the lowest temperature of 723 K and increased stress level. Very promising results were also achieved when parameter *C* was taken as stress-dependent, with the model error in the whole temperature range investigated being ~30%. However, good fit and low model error were obtained only for a linear fit function, which requires additional model analysis.

The model validation performed at 873 K and 80 MPa exposed the sensitivity of the model and correlation function algorithm and parameter *C* selection on the material and creep data available. In the case of 10CrMo9–10 steel and sustained stress levels (<100 MPa), accurate estimations of creep rupture times within 10% error were obtained when using the linear extrapolation function and parameter C = 18. Polynomial extrapolation function, however, resulted in more than 4–times overestimated values.

Evaluation results also emphasized that the fit adequacy parameter M cannot be used to predict whether or not a model is appropriate. However, it gives an insight into which model specifications are better than the others. The statistical calculations suggested that the polynomial function should give better response, but it turned out that the linear fitting was closer to experimental measurements.

## Figures and Tables

**Figure 1 materials-15-04431-f001:**
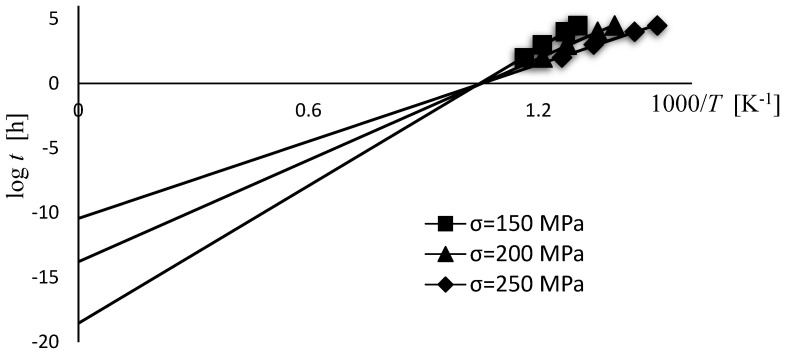
Example of iso−stress lines for 10CrMo9–10 steel [20] presented in a plot of (log *t*) vs. (1/*T*).

**Figure 2 materials-15-04431-f002:**
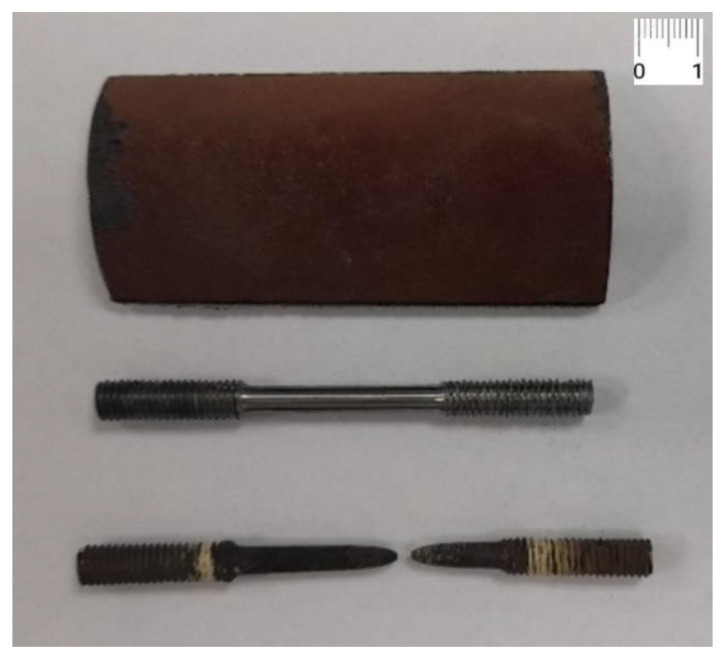
Sample of pipe made of 10CrMo9–10 steel (**above**), creep test specimen Type B 4 × 20 DIN 50125: 2016 (**middle**) and creep test specimen after creep testing (**below**).

**Figure 3 materials-15-04431-f003:**
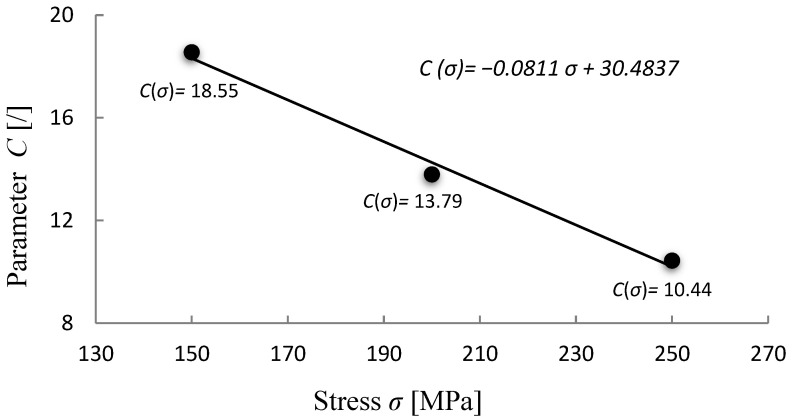
Linear correlation of C parameter for given 10CrMo9–10 steel data. The black points represent the values that come from the intersection of the iso–stress lines by the vertical axis y (1/*T* = 0) depending on the applied stress; see Figure 1.

**Figure 4 materials-15-04431-f004:**
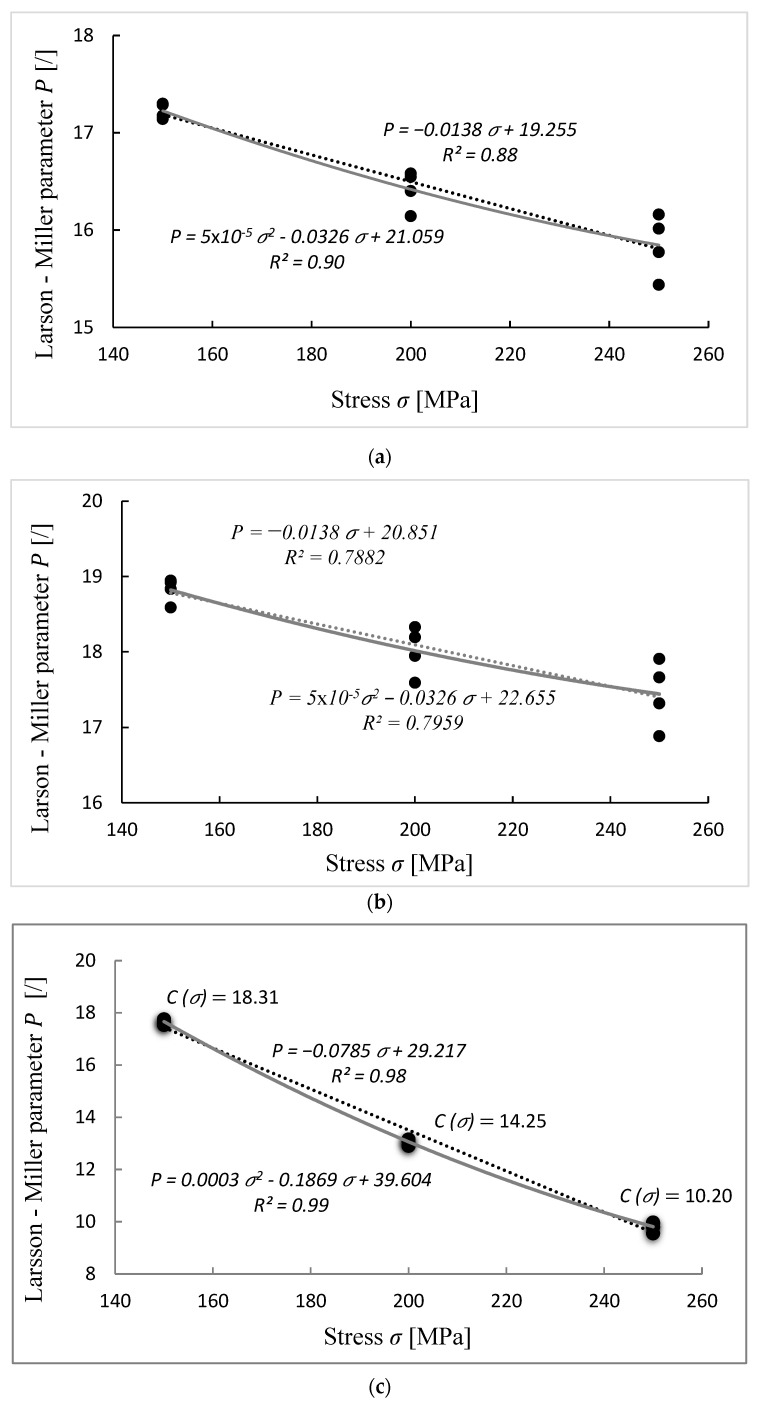
Chart of P vs. stress for different assumptions of parameter C and two extrapolation functions, linear and polynomial: (**a**) C=18, (**b**) C=20 and (**c**) C=f(σ). The black points represent the values of parameter P that were calculated according to Equation (5) (for (**a**,**b**)) and Equation (11) (for (**c**)), where measured data for temperature, stress and time to rupture (from the National Institute for Materials Science base) have been used.

**Figure 5 materials-15-04431-f005:**
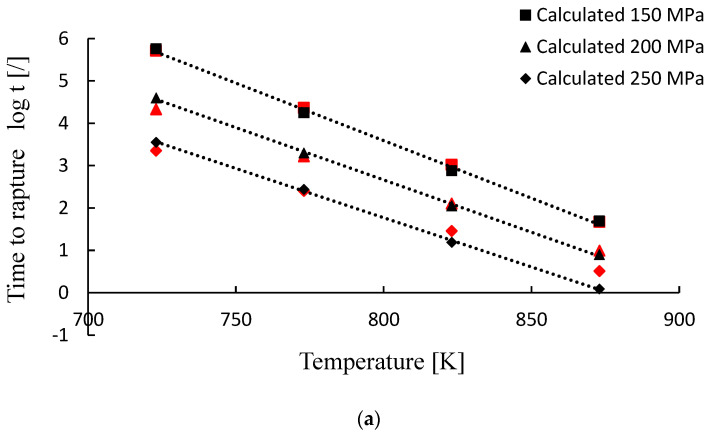
Creep-model performance compared with creep-test results. Model specifications are as follows: linear fit function (see Figure 4) and (**a**) C = 18, (**b**) C  = 20 and (**c**) C=f(σ) with C(150 MPa)  = 18.31, C(200 MPa)  = 14.25 and C(250 MPa)  = 10.20, respectively. Red points represent the measured values.

**Figure 6 materials-15-04431-f006:**
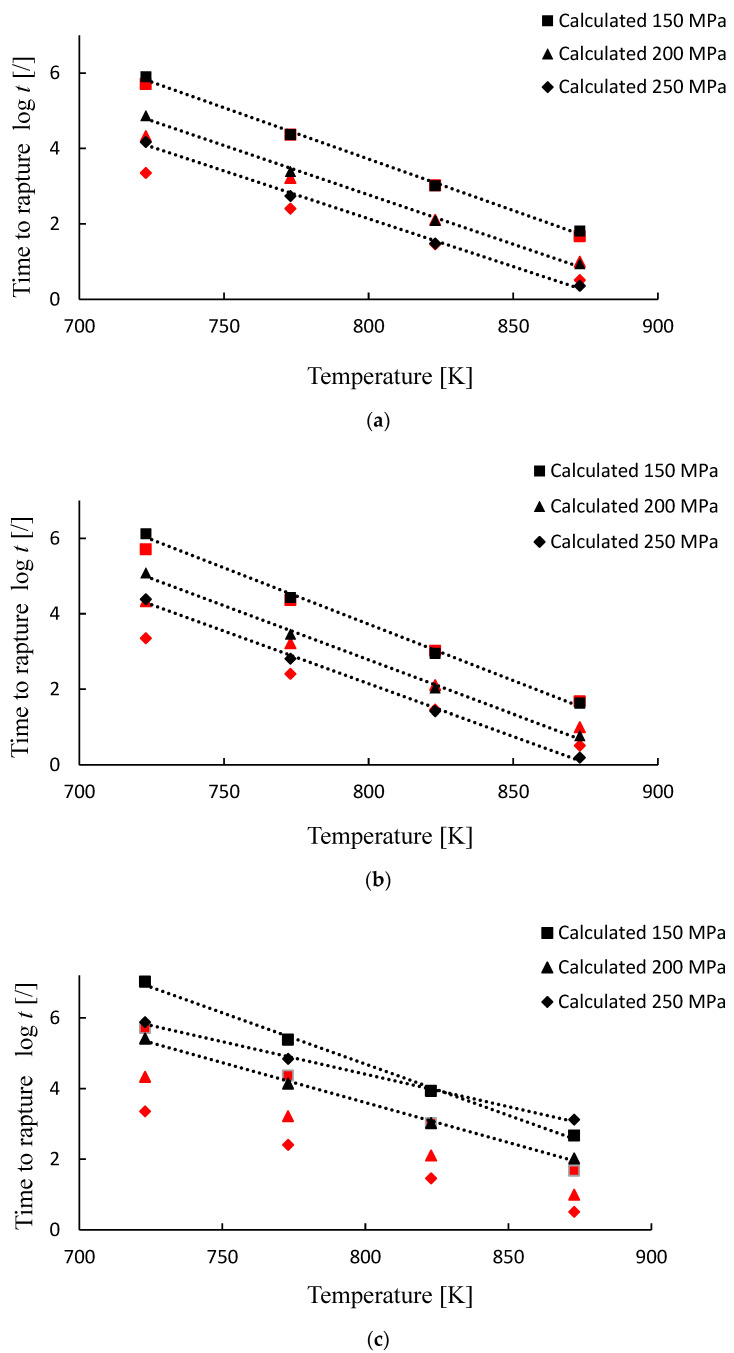
Creep-model performance compared with creep-test results. Model specifications are as follows: polynomial fit function (see Figure 4) and (**a**) C = 18, (**b**) C  = 20 and (**c**) C=f(σ) with C(150 MPa)  = 18.31, C(200 MPa)  = 1.25 and C(250 MPa)  = 10.20, respectively. Red points represent the measured values.

**Figure 7 materials-15-04431-f007:**
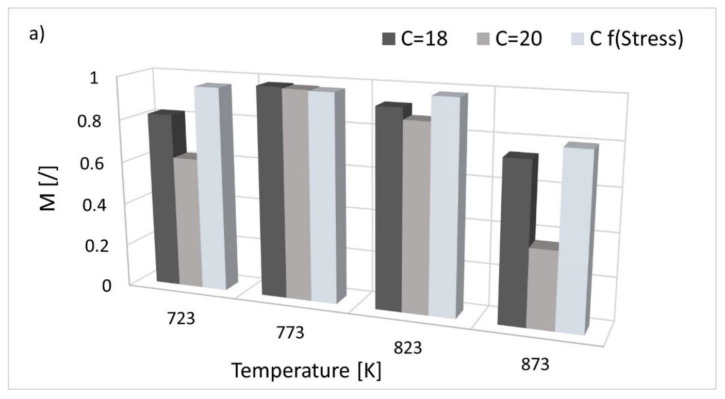
Comparison of fit adequacy parameter M for calculations performed using (**a**) the linear fit function and (**b**) the polynomial fit function, using different values of parameter C.

**Figure 8 materials-15-04431-f008:**
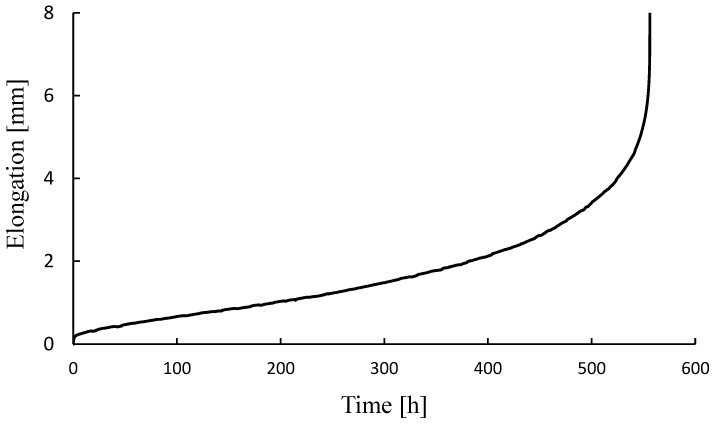
Creep rupture testing results for 10CrMo9–10 steel (T = 873 K, *σ* = 80 MPa).

**Table 1 materials-15-04431-t001:** P values extrapolated from the input creep data of 10CrMo9–10 steel.

*σ* [MPa]	Larson–Miller Parameter, P		
	C=18	C=20	C(*σ*); Equation (12)
Extrapolation by linear function			
150	17.19	18.79	17.44
200	16.50	18.09	13.51
250	15.80	17.40	9.59
Extrapolation by polynomial function			
150	17.29	18.89	18.31
200	16.54	18.13	14.22
250	16.03	17.63	11.62

**Table 2 materials-15-04431-t002:** The percentage error calculated when linear and polynomial fit functions were applied.

		Linear Extrapolation	Polynomial Extrapolation	Linear Extrapolation	Polynomial Extrapolation	Linear Extrapolation	Polynomial Extrapolation
		C = 18		C = 20		C (*σ*)	
*T* [K]	*σ* [MPa]	percentage error (%)		percentage error (%)		percentage error (%)	
723 K	150	−15	−62	−85	−161	−27	−1968
	200	−206	−252	−393	−468	−29	−1126
	250	−214	−567	−418	−976	+48	−339,910
773 K	150	+26	−2	+14	−18	+23	−953
	200	−32	−51	−53	−75	−3	−746
	250	−10	−117	−27	−152	+36	−27,330
823 K	150	+27	+1	+37	+14	+27	−746
	200	+14	+2	+25	+15	−15	−734
	250	+44	−6	+52	+8	+1	−29,497
873 K	150	−3	−36	+31	+8	+1	−897
	200	+21	+11	+46	+40	−71	−1003
	250	+61	+28	+74	+52	−89	−40,542

**Table 3 materials-15-04431-t003:** Estimated time to rupture at 80 MPa and 873 K for different assumptions of LMP model.

Extrapolation by Linear Function		
C = 18	C = 20	C(*σ*); Equation (12)
P = 18.15	P = 19.74	P = 22.93
618 h	416 h	192 h
Extrapolation by polynomial function		
C = 18	C = 20	C(*σ*); Equation (12)
P = 18.77	P = 20.36	P = 26.52
3174 h	2137 h	2,800,087 h

## Data Availability

Not applicable.

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
