# Peer review of "Creep Life Prediction of 10CrMo9–10 Steel by Larson–Miller Model"

_materials, 2022, doi:10.3390/ma15134431_

Round 1
Reviewer 1 Report
firstly, the title of the manuscript is too large and not sufficent explicite. It is necessary to indicate thr studed alloy. In fact only one alloy has been studied.
Secondly, it seems that the C parameter is a key point of this study. It is extremly important to define this parameter specificly, to explain the real physical meaning of this parameter, to discuter the influence of material characteristics (heat treatment, microstructure, mechancial properties detc...), test environment (temperature, loading, deformation mecanisms etc...) and to justify the chosen/used C parameter in this study.
Thirdlyly, in caption of figures or table, it is absolutely to precise the test condition (temperature, loading etc...). for example, in figure 1, 3, 4 and table 1,
Furthermore, some details should added to lead a better understanding:
- in figure 2, it is important to add a scale;
- in caption of figure 3 and figure 4, it is important to explain what correspond to the dark points;
- in figure 7, it is important to precise the temperature axis and the corresponding unity;
Reviewer 2 Report
The topic of this paper is related to using derived engineering parameters and models to predict and find the correlations between creep life (time to rupture), temperature, and stress. This research analyzes and compares different numerical algorithms using the Larson-Miller parameter (???) extrapolation model. The approach focuses on investigating the impact of different extrapolation methods and correlation functions to fit the LMP model. The numerical algorithm implemented is validated through the creep rupture test performed on 10CrMo9-10 steel at 873 K and 80 MPa. The simulation results turned out that only linear extrapolation function and ? taken as a constant value (other than taken as stress-dependent) provide adequate time to rupture prediction. This paper provides valuable methods, results, and solutions for determining the creep life of 10CrMo9-10 steel accurately. I recommend this paper can be revised to clarify the following issues:
1. In the Abstract part, line 6, the author stated, “under the effect of low stress.” Usually, the creep of material is considered a serviceability problem. However, the creep could happen under high sustained load as well (at least for concrete structures). So, my suggestion is to change the “low stress” to “persistent/sustained stress”, which is more accurate.
2. In Line 17, please add a reference first (explain it in line 68) or explain the unit Kelvin (relationship between Celsius and Kelvin) here the first time a new term appears in a paper. The author can add (600 ℃) after 873K because Kelvin is not commonly used by some researchers/readers.
3. In Line 29, the author stated, “under the effect of high-stresses,” which contradicts the definition of creep in line 6. Again, please revise to “persistent/sustained stresses.”
4. Page 3, Figure 1, the resolution of numbers and the coordinate axis are not the same. Please re-edit it in Microsoft software (PowerPoint/Excel) to ensure the Figure is of high quality.
5. In Lines 119 to 120, please explain each chemical element. For Example, “Cr-Chromium,” “Mo-Molybdenum,” etc.
6. In Line 124, please add the organization (National Institute for Materials Science) and country for this online database.
7. In Line 138, the authors perform the experiment with applied stress of 80 MPa. However, there are three different stress levels (150, 200, and 250 MPa) were selected as the creep tests before. Could the authors explain the inconsistency between these conditions?
8. In Line 149, the unit of time shown in this paper (tM, t) is in hours? If so, please describe it at the first time the term “t” appears.
9. Page 6, Figure 3. Please explain (1) What the data point (black circle marker and the shaded marker on the left of this marker) means and (2) what is the meaning of three short grey lines at C = 18.55, 13.79, and 10.33, respectively. Please also make sure all the numbers shown in one figure is unified (with the same font size and font type. Do not make some of the numbers bold and some of the numbers inclined.)
10. Page 6, line 195, please explain the definition of the coefficient of determination (R2).
11. Page 7, Figure 4(c), please explain what the data point (black circle marker and the shaded marker on the left of this marker) means.
12. Page 8, line 230, Chapter 2.1 of reference [18]. Please specify.
13. Page 9, line 242, please make a space before and after the sign “=”. Please also check all these problems shown in this paper and ensure all the formulas are uniform.
14. Page 10 to 11, Figures 5 and 6, the legend of these two figures only explains the black markers meaning. However, there are some red markers shown in these figures as well. Please explain what they are.
15. Page 12, Figure 7. (1) What is the “(/)” means after parameter M in the label y. (2) Please add x label name (Temperature) in this figure.
16. Page 13, Figure 8, the label y is wrong. Because the unit of strain should be (mm/mm) other than mm. This Figure’s y label looks like the deformation of the 20 mm length specimen. What is more, please also give the failure creep strain of the specimens because it is important for the creep tests (the different specimens should have different creep failure strains, please summarize them in one table.)
17. Page 14, line 538, “at low stress level.” The reviewer is curious about how to define the low, medium, and high stress levels in this paper. Please specify it here, not later. If possible, it is better to add a reference to explain this.
Finally, I will give two suggestions about all the figures shown in this paper. The resolution is also an essential factor in improving the paper acceptance rate for a high-quality paper. (1) When plotting a Figure and adding notation (words, numbers) on an original paper, it is better for the figure can be plotted in Microsoft Excel and edited in Microsoft software (PowerPoint or Word) and group them. Then, insert it into Microsoft Word. In this way, the resolution of the notation shown on each picture will have the same resolution as the words in this paper. Please also make sure the notations shown in each Figure have the same size (width and height). (2) Please check the minor edit and grammar errors shown in this paper.
Reviewer 3 Report
Title: Extrapolation of Creep Life Data by Larson-Miller Model
Manuscript ID: materials-1757583
Authors: Gustin et al.
Dear Authors,
Thank you for the opportunity to read your article. I found the topic is interesting and fundamental but can be practically useful. Generally speaking, there are some results presented in order to capture some trends, but the introduction, methods and results need more clear and deep explanation with fair point of view. I suggest that this article will be revised before its re-submission for another review process if applicable. As a conclusion, I recommend its major revision at this state.
I hope my comments are helpful.
Good luck,
A reviewer
Major concerns:
“Abstract”
-Lines 15-16: “…to choose the best extrapolation fit function and error tolerance.”->Please consider briefly stating your findings in this aspect.
-Line 18: “…rapidly…”->…significantly…?
-Lines 18-19: “…change the estimated time to rapture.”->Please consider stating the time.
“Keywords”
->Please consider providing keywords that are not used in the article title.
“1. Introduction”
-In Introduction, based on your literature review, (a) please consider clearly stating the research gap(s) you tried to address in this study. In other words, please consider explaining why you studied “the efficiency of the LMP model by matching the fitting material parameter C” and “the impact of different approaches in terms of extrapolation and correlation functions applied to fit the LMP equations.” (b) Please consider stating the unique contribution(s) of your study, in comparison with previous studies including [18,19].
“2. Creep modeling”
-Lines 130-131: “…150, 200 and 250 MPa…723 K to 823 K.”->Please consider providing your justification of these stress and temperature range you studied.
“2.3 Model validation”
-Lines 177-178: “It would be also interesting to verify other extrapolation functions, however is out of the scope of this investigation.”->Is this against what you wrote in the abstract “…to choose the best extrapolation fit function…” on line 16 ?
“4. Results and discussion”
-Lines 263-264: “Error is over 150% and increases with increased stress level”->Please consider stating the potential reason(s) for this error and how it can be reduced.
“5. Conclusions”
-In general, please focus on your study as this is your original article. Please find some detail comments below.
-Line 540: “…C is not one constant value for all materials…Larson and Miller…under all test’s conditions.”->Please consider minimizing the literature review contents in conclusions.
-Lines 570-571: “…however was not described in the article…”->Please consider discussing any contents first in the results and discussion (or other part of the article) and then providing a short summary in Conclusions.
Minor concerns:
-Please consider polishing English more. You may use some of my comments above for this purpose.
Round 2
Reviewer 1 Report
in the revised version of the manuscript, author has made necessery corrections including all remarks from reviewers.
Reviewer 3 Report
Dear Authors,
As all the comments were addressed, I would suggest the journal accept this article for its publication.
Best regards,
A reviewer